# Cytokine CCL9 Mediates Oncogenic KRAS-Induced Pancreatic Acinar-to-Ductal Metaplasia by Promoting Reactive Oxygen Species and Metalloproteinases

**DOI:** 10.3390/ijms25094726

**Published:** 2024-04-26

**Authors:** Geou-Yarh Liou, Crystal J. Byrd, Peter Storz, Justin K. Messex

**Affiliations:** 1Center for Cancer Research and Therapeutic Development, Clark Atlanta University, Atlanta, GA 30314, USA; 2Department of Biological Sciences, Clark Atlanta University, Atlanta, GA 30314, USA; 3Department of Cancer Biology, Mayo Clinic, Jacksonville, FL 32224, USA

**Keywords:** KRAS^G12D^, CCL9, PDAC initiation, ROS, MMP14, MMP3, ADM, pancreatic cancer

## Abstract

Pancreatic ductal adenocarcinoma (PDAC) can originate from acinar-to-ductal metaplasia (ADM). Pancreatic acini harboring oncogenic *Kras* mutations are transdifferentiated to a duct-like phenotype that further progresses to become pancreatic intraepithelial neoplasia (PanIN) lesions, giving rise to PDAC. Although ADM formation is frequently observed in Kras^G12D^ transgenic mouse models of PDAC, the exact mechanisms of how oncogenic Kras^G12D^ regulates this process remain an enigma. Herein, we revealed a new downstream target of oncogenic Kras, cytokine CCL9, during ADM formation. Higher levels of CCL9 and its receptors, CCR1 and CCR3, were detected in ADM regions of the pancreas in p48^cre^:Kras^G12D^ mice and human PDAC patients. Knockdown of CCL9 in Kras^G12D^-expressed pancreatic acini reduced Kras^G12D^-induced ADM in a 3D organoid culture system. Moreover, exogenously added recombinant CCL9 and overexpression of CCL9 in primary pancreatic acini induced pancreatic ADM. We also showed that, functioning as a downstream target of Kras^G12D^, CCL9 promoted pancreatic ADM through upregulation of the intracellular levels of reactive oxygen species (ROS) and metalloproteinases (MMPs), including MMP14, MMP3 and MMP2. Blockade of MMPs via its generic inhibitor GM6001 or knockdown of specific MMP such as MMP14 and MMP3 decreased CCL9-induced pancreatic ADM. In p48^cre^:Kras^G12D^ transgenic mice, blockade of CCL9 through its specific neutralizing antibody attenuated pancreatic ADM structures and PanIN lesion formation. Furthermore, it also diminished infiltrating macrophages and expression of MMP14, MMP3 and MMP2 in the ADM areas. Altogether, our results provide novel mechanistic insight into how oncogenic Kras enhances pancreatic ADM through its new downstream target molecule, CCL9, to initiate PDAC.

## 1. Introduction

Pancreatic ductal adenocarcinoma (PDAC) remains one of the most challenging and most fatal types of cancer. With an extremely low survival rate and no significant improvements in almost five decades, the current five-year survival rate is an abysmal 12% for PDAC patients, which is far lower than other leading cancer sites such as the breast, colon, prostate and lungs [1]. The high death toll of PDAC is the result of insufficient examination tools that can detect PDAC in its early stage, as well as a lack of effective therapeutic treatment options [2,3,4,5,6].

PDAC originates from pancreatic acinar cells whose major function is to synthesize and secrete digestive enzymes for breaking down food. In the presence of stimuli such as inflammation and growth factors [7,8,9,10,11,12], quiescent acini undergo an acinar-to-ductal metaplasia (ADM) process to become proliferative duct-like cells with progenitor features [10,13,14,15]. Clinical data demonstrate that approximately 95% of PDAC patients possess oncogenic *Kras* mutations including *Kras^G12D^* (dominantly) and *Kras^G12V^* [16,17,18]. Accumulative evidence from linear-tracking studies in genetically engineered mice as well as in vitro 3D organoid culture systems shows that oncogenic *Kras* mutations, when expressed in pancreatic acini, transdifferentiate these cells to a duct-like phenotype which can further develop into PDAC, along with other gene mutations [19,20,21,22]. Furthermore, when Kras^G12D^ was specifically turned off in the acini of mouse pancreas in a GEMM, it reversed PDAC cells back to acini [23], revealing the mastermind role of oncogenic Kras^G12D^ in PDAC. Other studies also unveiled that Kras^G12D^ is required to drive the immune evasion of PDAC [24,25,26]. Moreover, it also has been shown that Kras^G12D^ recruited inflammatory macrophages to accelerate PDAC initiation through promoting ADM [27].

Since there are no inhibitors for oncogenic Kras mutant proteins, such as Kras^G12D^ and Kras^G12V^, and given their importance in PDAC’s initiation and progression, a comprehensive understanding of their downstream targets and their effects on pancreatic ADM during PDAC initiation is pivotal to developing new strategies for the early detection and intervention of PDAC. Although ADM formation is frequently observed in Kras^G12D^ transgenic mouse models of PDAC, the exact mechanisms of how oncogenic Kras^G12D^ regulates this process remain an enigma. In this study, we identified cytokine CCL9 as a new downstream target of oncogenic Kras^G12D^ to promote PDAC initiation through ADM. It is noted that CCL9 is also known by various names such as macrophage inflammatory protein-1 gamma (MIP-1ɣ), macrophage inflammatory protein-related protein-2 (MRP-2) and CCF18 in rodents. We showed that the knockdown of CCL9 in Kras^G12D^-expressed acini reduced Kras^G12D^-induced ADM in a 3D organoid culture system. In addition, exogenously added CCL9 or overexpression of CCL9 was capable of driving pancreatic ADM. Upregulation of reactive oxygen species (ROS) through the NAPDH oxidase system and metalloproteinases (MMPs) including MMP14, MMP3 and MMP2 were demonstrated as downstream mediators of CCL9 to modulate pancreatic ADM. In transgenic mice of p48^cre^:Kras^G12D^, depletion of CCL9 using its specific neutralizing antibody abated ADM formation, pancreatic intraepithelial neoplasia (PanIN) structures and expression levels of MMP14, MMP3 and MMP2. Hence, our results revealed a novel mechanistic insight into how oncogenic Kras^G12D^ expedites pancreatic ADM through a new downstream target molecule, CCL9.

## 2. Results

### 2.1. Upregulation of Cytokine CCL9 and Its Receptors in the Pancreatic ADM Regions

Acinar-to-ductal metaplasia (ADM) is the process that oncogenic Kras^G12D^ utilizes to convert quiescent pancreatic acini to a proliferative duct-like phenotype, which initiates pancreatic ductal adenocarcinoma (PDAC). Results from the microarray experiments that were performed to identify the downstream targets of Kras^G12D^ during pancreatic ADM revealed that CCL9 is a top-ranked candidate. To verify this finding, we first evaluated the expression of cytokine CCL9 in ADM regions of p48^cre^:Kras^G12D^ mouse pancreas via immunohistochemistry (Figure 1A). The pancreas tissue from a littermate with a matching age was used as a control. As shown in Figure 1A, elevated expression of CCL9 was detected not only in the Kras^G12D^-induced ADM structures but also in the infiltrating immune cells surrounding the ADM regions. Our previous studies showed that macrophages are essential for pancreatic ADM formation and that one of the mechanisms that Kras^G12D^ utilizes to potentiate this pancreatic ADM process is by recruiting more macrophages to the local pancreas environment [11,27]. To test if the CCL9-poistive infiltrating immune cells are macrophages, we immunofluorescently labeled the pancreas tissue samples from the p48^cre^:Kras^G12D^ mouse and its control mouse p48^cre^ with antibodies of F4/80, which is a macrophage marker; CK-19, which is a ductal-marker; and CCL9 (Figure 1B,C). As shown in Figure 1B, some of the macrophages surrounding the ADM area were CCL9-positive, suggesting that some of the infiltrating macrophages expressed the CCL9 cytokine. The receptors for cytokine CCL9 include CCR1 and CCR3 [28,29,30,31]. Both were found upregulated in the Kras^G12D^-mediated pancreatic ADM (Figure 1D). Furthermore, real time qRT-PCR results from our 3D organoid culture system in which primary acini isolated from LSL-Kras^G12D^ mouse pancreas was induced to undergo ADM process by expressing Kras^G12D^ in the acini using an adenovirus delivery method [32], indicated increased mRNA levels of CCL9 and CCR3 (Appendix A). Altogether, these data suggest that Kras^G12D^ upregulated CCL9 and its receptors during ADM.

It is well-known that approximately 95% of human PDAC patients harbor oncogenic Kras mutations [16,17,18]. Therefore, we assessed the levels of CCL15, a human orthologue of mouse CCL9, and its receptors in human pancreatic ADM regions. As shown in Figure 1E,F, an elevated expression of CCL15 and its receptors CCR1 and CCR3 was present in human ADM.

### 2.2. CCL9 Was Required for Oncogenic Kras^G12D^-Induced ADM of the Pancreas and a New Pancreatic ADM Driver

To determine whether or not CCL9 functions as a downstream target of oncogenic Kras^G12D^ to regulate ADM during PDAC initiation, we knocked down *CCL9* using shCCL9 lentivirus in primary pancreatic acini that were infected with ad-cre to induce Kras^G12D^ expression and then evaluated the acinar-to-ductal metaplasia ability of these acini using our 3D organoid culture system. The expression of oncogenic Kras^G12D^ in the acini increased pancreatic ADM events 7–8-fold (Figure 2A). Furthermore, this Kras^G12D^-induced ADM was dramatically reduced by knocking down *CCL9* via shCCL9 lentivirus (Appendix A), suggesting that CCL9 was required for the Kras^G12D^-induced pancreatic ADM process.

If CCL9 was truly downstream of oncogenic Kras^G12D^ in the process of ADM, CCL9 would be sufficient to drive pancreatic ADM. To further bolster our results obtained from Figure 2A, we first treated primary acinar cells that harbored wildtype Kras with recombinant murine CCL9, followed by embedding them in collagen within a 3D setting to test their ability to become duct-like structures. As shown in Figure 2B, exogenous addition of CCL9 resulted in the transdifferentiation of acini to a duct-like phenotype. In addition, expression of *CCL9* in primary pancreatic acini through ad-CCL9 adenovirus also converted acinar cells to proliferating ductal-like cells (Figure 2C). Besides the change in morphology due to a conversion of two different cell types containing acini and ducts during ADM, the results of real-time qRT-PCR also showed a reduction in Mist-1 transcripts and an elevation in cytokeratin 19 (CK-19) and mucin-1 transcripts in these duct-like cells (Figure 2D). These results were consistent with all previous reports [33,34,35] and suggested that these cells underwent acinar-to-ductal metaplasia. Altogether, these data not only indicate that CCL9 functions downstream of Kras^G12D^ to induce ADM but also reveal that CCL9 is a new player in driving the pancreatic ADM process.

### 2.3. CCL9 Promoted Pancreatic ADM through Regulation of Reactive Oxygen Species (ROS)

Reactive oxygen species (ROS) have been reported to participate in pancreatic ADM [36,37]. In addition, cytokine CCL9 is also called macrophage inflammatory protein-1 gamma (MIP-1ɣ), which suggests that CCL9 is likely to regulate oncogenic Kras-induced pancreatic ADM via ROS levels. To test this possibility, we first evaluated the intracellular ROS levels of the primary pancreatic acini that had expressed Kras^G12D^ with *CCL9* knockdown. As shown in Figure 3A, overexpression of Kras^G12D^ increased levels of total intracellular ROS in primary acini. Furthermore, knockdown of *CCL9* using shCCL9 lentivirus was able to significantly diminish Kras^G12D^-induced intracellular ROS levels, suggesting that Kras^G12D^ signaled to CCL9 to modulate ROS. Indeed, when overexpressing CCL9 alone in primary murine pancreatic acini that harbored wildtype *Kras* (Appendix A), intracellular ROS levels dramatically increased up to 4-fold in comparison with the control cells (Figure 3B). We next assessed the effect of depletion of ROS in pancreatic ADM in response to CCL9 induction in our 3D organoid culture system. As shown in Figure 3C, depletion of ROS using the general ROS scavenger, N-acetyl-L-cysteine (NAC) [38,39,40], almost completely abolished CCL9-induced acinar-to-ductal metaplasia in the 3D organoid culture. Meanwhile, depletion of ROS through NAC had no effect on impacting cell viability (Figure 3D). To further dissect the source of intracellular ROS generated by cytokine CCL9, we knocked down *p22phox*, an essential subunit of the NADPH oxidase system, in primary pancreatic acini that expressed CCL9 to determine the cellular ROS levels of these cells. As shown in Figure 3E, knockdown of *p22phox* completely reduced ROS production even below the basal levels, suggesting that CCL9-induced ROS is mainly produced through NADPH oxidase system. Furthermore, knockdown of *p22phox* completely blocked CCL9-induced ADM of the pancreas (Figure 3F).

### 2.4. CCL9 Upregulated Metalloproteinases (MMPs) including MMP14, MMP3 and MMP2 to Modulate Pancreatic ADM

The ADM process involves converting quiescent acini to proliferating ductal cells, which requires remodeling the extracellular matrices (ECMs) via MMPs. In the ADM regions of p48^cre^:Kras^G12D^ mouse pancreas, we observed a higher expression of MMP3 and MMP14 (Figure 4A). In addition, a slight increase in MMP2 around ADM regions, especially on the surrounding infiltrating immune cells, was also detected (Appendix A). In human pancreas tissues, as shown in Figure 4B,C, an elevated expression of both MMP3 and MMP14 was also present, specifically in the ADM areas, as judged by the ductal marker CK-19 expression as compared to the normal pancreas. We next utilized our 3D organoid culture system to evaluate if the increased levels of MMP proteins in ADM were due to an upregulation of MMP gene expressions by Kras^G12D^ and/or CCL9. As shown in Figure 4D, the results of the real-time qRT-PCR indicated increased mRNA levels of MMP2, MMP3 and MMP14 in the Kras^G12D^-expressed acini of the pancreas that underwent ADM in 3D organoid culture. The same result was obtained from the pancreatic acini expressing CCL9 (Figure 4E). Furthermore, a higher activity of MMP2 and MMP3 was detected in the conditioned media of ad-CCL9 pancreatic organoid culture (Figure 4F), suggesting that these CCL9-upregulated MMPs were enzymatically active. To further dissect the relationship among Kras^G12D^, CCL9 and MMPs, we treated the CCL9-expressing pancreatic acinar cells with GM6001, a generic inhibitor of MMPs in our 3D organoid culture system for their ability to form the duct-like structures. As shown in Figure 4G, blockade of MMPs via GM6001 treatment not only just completely abolished CCL9-induced ADM, but further reduced the number of ADM events below the basal level, thus supporting our previous results in Figure 4B,C,E.

Given that, so far, the specific inhibitor targeting each individual MMP remains unavailable, we utilized shMMP14 and shMMP3 lentivirus to specifically knock down either *MMP14* or *MMP3* to define their role in CCL9-induced ADM as CCL9 downstream targets. When knockdown of *MMP14* was performed with two individual sequences through the lentivirus delivery system, either of them significantly decreased CCL9-induced ADM in our 3D organoid culture system (Figure 5A). Furthermore, an ectopic expression of MMP14 in primary murine acinar cells of the pancreas was capable of driving ADM (Figure 5B). Similarly, knockdown of *MMP3* by shMMP3 lentivirus significantly reduced CCL9-caused pancreatic ADM (Figure 5C), and the overexpression of MMP3 promoted pancreatic ADM formation (Figure 5D). Notably, MMP14 was a more potent ADM driver than MMP3 in the 3D organoid culture system. Altogether, our results suggested a new signaling axis of Kras^G12D^/CCL9/MMP14 and MMP3 to initiate pancreatic ductal adenocarcinoma through promoting ADM in the pancreas.

### 2.5. Blockade of CCL9 Diminished Oncogenic Kras^G12D^-Mediated Pancreatic ADM and Reduced Local Inflammation and MMPs

To investigate the effect of CCL9 inhibition in oncogenic Kras-regulated PDAC initiation, we treated p48^cre^:Kras^G12D^ mice and their age-matched littermates with a specific CCL9 neutralizing antibody (CCL9 NAb) for 5 weeks. CCL9 NAb treatment reduced Kras^G12D^-mediated abnormal pancreatic structures, including ADM and PanIN lesions, as shown in the H&E stain (Figure 6A). A further detailed analysis of these structures indicated a 56% decrease in ADM and a 53% decrease in PanIN 1 lesions (Figure 6A). Our previous studies have shown a critical role of infiltrating macrophages in PDAC initiation [11,27,41]. To test if blockade of CCL9 also affects infiltrating immune cells, especially macrophages, in Kras^G12D^-mediated PDAC initiation, we examined the infiltrating immune cells, including macrophages, T cells and neutrophils, in CCL9 NAb-treated pancreas tissues of p48^cre^:Kras^G12D^ mice using immunohistochemistry (IHC) (Appendix A). A subsequent quantitative IHC analysis indicated CCL9 NAb significantly decreased infiltrating macrophages nearby the ADM regions (Figure 6B) instead of T cells and neutrophils (Figure 6C,D). Next, we evaluated if CCL9 NAb also impacted the levels of MMP14, MMP3 or MMP2, all of which were identified in our 3D organoid culture system (Figure 4E). As shown in Figure 6E and Appendix A, MMP14 was expressed in the ADM and surrounding immune cells of the pancreas in p48^cre^:Kras^G12D^ mice. Treatment with CCL9 NAb significantly diminished the expression of MMP14 in Kras^G12D^-mediated pancreatic ADM regions. Similarly, CCL9 NAb treatment also reduced MMP3 and MMP2 levels in the ADM of the pancreas (Figure 6F,G).

## 3. Discussion

Oncogenic Kras mutations such as *Kras^G12D^* were found in more than 95% of PDAC patients. Furthermore, evidence from transgenic mouse studies demonstrated that oncogenic *Kras^G12D^* mutation is a necessity during PDAC initiation [23,42,43], progression and dissemination [44,45,46]. Besides *Kras^G12D^*, the loss of tumor suppressor genes including *Smad4*, *Ink4a*, *Lkb1*, *p53*, etc., is also essential for the progression and dissemination of PDAC [44,47,48,49,50,51]. Although the Kras^G12C^ inhibitor, known as adagrasib, has been developed and has been FDA-approved since December of 2022 for treating lung cancer that harbors this mutation, an inhibitor of Kras^G12D^ remains absent. Because of the essential role of oncogenic Kras^G12D^ in all stages of PDAC development, including initiation, progression and metastasis, a comprehensive study on the downstream targets of Kras^G12D^ is necessary for developing new strategies for effective intervention in the Kras^G12D^ signaling of PDAC.

In this current study, we identified cytokine CCL9 as a new downstream target of Kras^G12D^ to modulate pancreatic ADM during PDAC initiation (Figure 2). In addition to its involvement in asthma severity [52], CCL15, a human orthologue of mouse CCL9, has also been reported to mediate the recruitment of suppressive monocytes to liver cancer and subsequently facilitate metastatic abilities of liver cancer cells [53]. Similarly, in colon cancer, cancers that lack SMAD4 expressed more CCL15, which attracted CCR1^+^ myeloid cells, leading to enhanced liver metastasis [54,55]. Other groups also reported that in addition to myeloid cells, these colon cancer cells were also able to recruit CCR1^+^ tumor-associated neutrophils to facilitate lung metastasis through the expression of CCL15 [56]. While these reports indicated the role of CCL15 in colon cancer-caused metastasis, our previous work also demonstrated that CCL15 elevated PDAC cell migration and invasion in human-cultured PDAC cell lines that possess the Kras^G12D^ mutation [46]. Regarding the involvement of CCL9 in pancreatic cancer and pancreatitis, in addition to our results shown here and as mentioned previously [46], another recent study has reported that pancreatic cancer cells of Pdx-1^cre^:Kras^G12D^:TP53^R172H^ (KPC) mice upregulated CCL9 to recruit granulocytic myeloid-derived suppressor (G-MDSC) cells as an alternative mechanism to resist to drug-induced cell death, even when a higher CD8^+^ T cell response was induced by drug treatment [57]. In a cerulein-induced pancreatitis mouse model, Fang et al. identified 18 genes comprising the acute pancreatitis-specific gene signatures in the immune microenvironment via scRNA-seq and bulk RNA-Seq technologies, and CCL9 was among these 18 genes [58]. In addition, Sakuma et al. identified chemokine CXCL16 as an acute pancreatitis maker from patient serum samples, and demonstrated that the CXCL16 knockout mice had attenuated pancreatitis-induced acini necrosis due to the decreased expression of CCL9, which attracted neutrophil infiltration in the pancreas [59].

Our results showed that CCL9 induced pancreatic ADM via the increase in the intracellular levels of ROS (Figure 3). Although several pro-inflammatory cytokines such as TNF, IFN-ɣ and IL-1β have been shown to promote cellular ROS through mitochondria and NAPDH oxidase in retinal epithelial cells [60], so far, the exact mechanism of how the pro-inflammatory cytokine CCL9 potentiates intracellular ROS remains largely unknown. Our previous study showed that the human orthologue of mouse CCL9, CCL15, promoted ROS to mediate the migration ability of cancer cells via p22phox, which is an essential subunit for the formation of functionally active NADPH oxidase [61], thus suggesting CCL9 can at least modulate the levels of intracellular ROS through NADPH oxidase [46]. Indeed, our results demonstrated that CCL9 produced ROS via the NADPH oxidase system, as when we knocked down the essential subunit of the system, p22phox, it completely abolished CCL9-mediated intracellular ROS (Figure 3E) as well as causing pancreatic ADM (Figure 3F).

We demonstrated that during pancreatic ADM, an increased expression of CCL9 was not only detected in pancreatic epithelial cells but also present in the surrounding immune cells such as the recruited macrophages (Figure 1A,B). This is also the case in regard to its receptors, including CCR1 and CCR3 (Figure 1D). These results suggested two mechanisms utilized by Kras^G12D^-expressed pancreatic acini to initiate PDAC through the upregulation of ADM. One is the autocrine mechanism, in which Kras^G12D^ acini were able to elevate both CCL9 and its receptors CCR1 and CCR3 by themselves, and the other is through the paracrine mechanism, in which Kras^G12D^ acini expressed CCR1 and CCR3 to attract CCL9^+^ macrophages. This reveals a new regulating system between Kras^G12D^ pancreatic acini and infiltrating macrophages. This system exploits the interaction between CCL9 and CCR1/CCR3, as the previous report indicated that Kras^G12D^ acini of the pancreas expressed ICAM-1, to recruit inflammatory macrophages for accelerating PDAC initiation [27].

The tumor microenvironment plays a pivotal role in determining cancer cell fate. It has been reported that several subgroups of macrophages express different protein markers during PDAC initiation and progression to help facilitate pancreatic cancer growth and survival [62,63,64,65,66]. These proteins expressed on macrophages include TNF [11], RANTES [11], phospho-STAT1 [25], CD163 [25] and IRF4 [67], and expression of these protein markers render macrophages either more inflammatory, which expediates ADM during PDAC initiation, or more immune-suppressive, which is required for the precursors of PDAC, PanIN lesions, to grow during PDAC progression. Among these subpopulations, TNF^+^ and/or RANTES^+^ macrophages have been shown to promote PDAC initiation in Kras^G12D^ pancreas via increases in the frequency of pancreatic ADM events and the size of the derived duct-like cells, thus accelerating pancreatic ADM of Kras^G12D^ acini [11,27]. However, whether these TNF^+^ and/or RANTES^+^ macrophages also express CCL9 or are composed of different subpopulations altogether during the ADM process remains unknown. If CCL9^+^ macrophages are a new subpopulation of the infiltrating macrophages in the local environment during PDAC initiation, determination of their potency in comparison to other subpopulations on pancreatic ADM induction are currently under investigation.

Matrix metalloproteinases (MMPs) are crucial to the regulation of the behavior as well as architecture of cells, as these enzymes can not only degrade the extracellular matrix (ECM) but can also process numerous molecules such as growth factors, cell surface receptors, cell–cell adhesion molecules, etc. [68,69]. It has been shown that pancreatic acini from MMP7 knockout mice failed to undergo ADM in a pancreatitis model as well as in response to EGF activation [70,71]. In addition to MMP7, an upregulation of MMP9 transcripts was detected in transdifferentiated duct-like cells of ADM when expression of NF-κB, an ADM driving gene, was high in primary pancreatic acini [11]. Interestingly, when expressing Kras^G12D^ or CCL9, each of which induces pancreatic ADM, our data showed an increase in mRNA levels of MMP2, MMP3 and MMP14 (Figure 4D,E). In addition, an elevated expression of these MMP proteins were also detected in the ADM regions of p48^cre^:Kras^G12D^ mouse pancreas as well as human pancreas tissues (Figure 4A,B and Figure 6E–G), suggesting these newly identified MMPs are also involved in modulating the pancreatic ADM process. Indeed, ectopically expressed MMP14 or MMP3 in primary pancreatic acini was capable of driving ADM formation; however, the MMP14-induced ADM effect was much higher compared to that induced by MMP3 (Figure 4F,G). Altogether, these results, combined with previous reports from other MMPs including MMP7 and MMP9, suggest a potential beneficial use of the broad MMP inhibitor on high-risk populations of PDAC by mitigating pancreatic ADM events during PDAC initiation.

It has been reported that *CCL15*, the human orthologue of mouse CCL9, is an NF-κB target gene [72]. We also showed that when treating Kras^G12D^-expressed pancreatic acini with the specific NF-κB inhibitor BMS345541, it suppressed CCL9 transcription as well as abolished Kras^G12D^-induced pancreatic ADM (Appendix A). Altogether, this affirms that Kras^G12D^ can upregulate CCL9 via transcription factor NF-κB as one of its mechanisms.

Overall, our current study demonstrated that cytokine CCL9 is a new driver of pancreatic ADM and a new downstream target of oncogenic Kras^G12D^ during PDAC initiation. In addition, we also elucidated the mechanism utilized by CCL9 to transdifferentiate acini of the pancreas into a duct-like phenotype that can give rise to PDAC precursor lesions, PanIN. This is mediated through elevated levels of ROS as well as MMP proteins including MMP2, MMP3 and MMP14 (Figure 7). Given that CCL9 is a secreted factor that can be detected in the blood stream, the increased levels of serum CCL9 are not exclusively found in the early events of PDAC initiation. Other disorders such as pancreatitis can also cause elevated levels of serum CCL9. Future studies would be required to identify other secreted factors during pancreatic ADM and/or PanIN formation, allowing the formation of a specific panel constituting multiple identified markers for easy and regular early detection of PDAC in serum samples in the future.

## 4. Materials and Methods

### 4.1. Antibodies and Reagents

CCL9, CCL15, CCR3, MMP14 and CD3 antibodies were purchased from Abcam (Cambridge, UK). Both F4/80 and Ly6B.2 antibodies were from Bio-Rad (Hercules, CA, USA). CCR1 antibody was purchased from Novus Biologicals (Littleton, CO, USA). MMP2 and MMP3 antibodies were obtained from Thermo Fisher Scientific (Walthem, MA, USA). In addition, MMP3 antibody was also purchased from ProteinTech (San Diego, CA, USA). Recombinant CCL9 was from PeproTech (Cranbury, NJ, USA). CCL9 neutralizing antibody and its isotype control antibody were purchase from R&D Systems (Minneapolis, MN, USA) and BioLegend (San Diego, CA, USA), respectively. N-Acetyl Cysteine was purchased from Sigma-Aldrich (St. Louis, MO, USA). Other reagents used are described in the specific experiment sections.

### 4.2. Isolation and Cell Culture of Primary Pancreatic Acini

These procedures have been previously described in detail [11,27,36]. In brief, the minced mouse pancreas was incubated with collagenase I (1 mg/mL) in HBSS media at 37 °C and shaken for 20 min. Equal volume of ice-cold 5% FBS-HBSS media was added to stop the collagenase digestion, followed by the repeated step of centrifugation at 2000 rpm for 2 min (4 °C) and subsequent washing with ice-cold 5% FBS-HBSS media two times. The cell pellet containing pancreatic acini was resuspended in ice-cold 5% FBS-HBSS media, and the cell solution was pipetted through a sterile 500 µm mesh and then a sterile 105 µm mesh. The flow-through cells were resuspended in 30% FBS-HBSS in a drop-wise manner, and then spun down at 1000 rpm for 2 min (room temperature). Cells were resuspended in Waymouth media containing 1% FBS, 100 µg/mL trypsin inhibitor, 1 µg/mL dexamethasone and 100 U/mL penicillin/streptomycin (Waymouth complete media). Cells were cultured in a humidified 37 °C cell incubator with 5% CO_2_.

### 4.3. Acinar-to-Ductal Metaplasia Assay in 3D

The detailed procedure has been previously described [11,27,36]. In summary, cell culture well plates were coated with collagen I/Waymouth media. The freshly isolated primary pancreatic acini were mixed with collagen I and then added to the plate. The mixture of the cell and collagen I gel was overlayed with Waymouth complete media as indicated in the “Isolation and Cell Culture of Primary Pancreatic Acini” section. The culture media were replaced every two days. For recombinant CCL9 treatment, 50 ng/mL recombinant CCL9 was added to both the cell/gel mixture and the culture media. To scavenge reactive oxygen species (ROS), 5 mM NAC was added to the cell/gel mixture and media. The number of ductal cells, as judged by the change in morphology per well at day 5, was counted under a microscope, and the cell structures were photographed.

### 4.4. Gene Knockdown and Overexpression Using a Viral Delivery System

Lentiviral plasmids encoding shCCL9 and shScramble were purchased from Sigma-Aldrich (St. Louis, MO, USA) with the specific TRC clone ID numbers described in our previous publication [46]. Lentivirus was produced in 293FT cells using the ViraPower Packaging Mix (Invitrogen; Waltham, MA, USA) according to the manufacturer’s instruction. In addition, the lentiviral titers were determined based on the manual of the manufacturer (Invitrogen). Lentiviruses of shMMP3 and shMMP14 were purchased from Sigma-Aldrich with the following TRC clone ID numbers. For shMMP14, TRCN0000031264, clone ID NM_008608, and TRCN0000031264, clone ID NM_008608. For shMMP3, TRCN0000335079, clone ID NM_010809, and TRCN0000335080, clone ID NM_010809. To knockdown the indicated gene including CCL9, MMP14 and MMP3 and p22phox, cells were infected with shCCL9, shMMP14, shMMP3, p22phox or shScramble lentiviruses in the presence of 6 µg/mL fresh polybrene.

Adenoviruses, including ad-CCL9, ad-MMP3, ad-cre, ad-null and ad-MMP14, were purchased from Vector Biolabs (Malvern, PA, USA) and Charles River (Rockville, MD, USA), respectively. Adenovirus titers were determined using an Adeno-X rapid titer kit (BD Biosciences; San Jose, CA, USA) according to the manufacturer’s instruction. These adenoviruses were used to express genes of interest in the primary murine acini of the pancreas using the previously described method [32,37].

### 4.5. RNA Isolation and Real-Time Quantitative RT-PCR

Total RNA was isolated from cells using the RNeasy kit from Qiagen (Hilden, Germany) according to the manufacturer’s instruction. Levels of mRNA of interest were examined using a 2-step quantitative reverse transcriptase-mediated real-time PCR (qPCR) method. An equal amount of total RNA was converted to cDNA by the high-capacity cDNA reverse transcriptase kit (Applied Biosystems, Bedford, MA, USA). Real-Time quantitative PCR was performed in a CFX Connect real-time PCR detection system (Bio-Rad) using the TaqMan Universal PCR master mix (Applied Biosystems) with probe/primer sets and the following PCR program: 95 °C for 20 s; 40 cycles of 95 °C for one second; and 60 °C for 20 s. All Taqman gene expression assays were purchased from Applied Biosystems (Mist-1: Mm00627532_s1; CK-19: Mm00492980_m1; mucin-1: Mm00449604_m1; CCL9: Mm00441260_m1; MMP2: Mm00439498_m1; MMP3: Mm00440295_m1; MMP14: Mm00485054_m1; GAPDH: Mm99999915_g1 and 18s rRNA: Hs99999901_s1). The collected data were analyzed by Sequence Detection System software v3 (Bio-Rad) and normalized to the reference gene including GAPDH and/or 18s rRNA. The mRNA abundance was calculated using the ΔΔ*C*_T_ method.

### 4.6. Mouse Strains and Treatment

p48^cre^ and LSL-Kras^G12D^ mice were purchased from the Jackson Laboratory (Bar Harbor, ME, USA), and genotyped as previously described [25,37]. To neutralize CCL9 in vivo, 3-week-old p48^Cre^:Kras^G12D^ mice and their matching littermates were intraperitoneally injected with CCL9 neutralizing antibody or the isotype control antibody at a dose of 200 µg/kg every other day for 5 weeks. The mouse pancreas tissues were harvested at an age of 8 weeks. The sex of the mice was randomized in the experiments. All animal experiments were carried out under the animal protocol approved by the Atlanta University Center (AUC) IACUC, as well as in accordance with institutional and national guidelines and regulations.

### 4.7. Human Pancreas Tissue Samples

De-identified human pancreas tissue samples were obtained from US Biomax (Derwood, MD, USA), BioChain (Newark, CA, USA), Cooperative Human Tissue Network/CHTN (Rockville, MD, USA) and the Rapid Autopsy Program at University of Nebraska Medical Center (Omaha, NE, USA). All experiments involved in using human tissue samples are carried out according to the IRB protocol approved by the CAU IRB committee.

### 4.8. Immunohistochemistry

The procedure has been described in our previous publication [46]. In brief, slides were deparaffinized in xylene and gradually re-hydrated through 100% alcohol to distilled water. The rehydrated slides were subjected to heat-induced antigen retrieval in either 10 mM sodium citrate buffer (pH 6.0) or 10 mM Tris buffer (pH 9). After treating with 3% hydrogen peroxide followed by PBS wash, slides were incubated with protein block serum free solution (Agilent DAKO; Santa Clara, CA, USA) for 10 min at room temperature. After the primary antibody was incubated on the slides, the ImmPRESS Polymer Detection Kit (Vector Laboratories; Burlingame, CA, USA) was used according to the manufacturer’s instructions. For immunofluorescent staining, instead of using the ImmPRESS Polymer Detection Kit, Alexa-Fluor-conjugated secondary antibodies (Thermo Fisher Scientific) corresponding to the primary antibody species were applied to the slides according to the manufacturer’s instructions. Images were collected using the Aperio VERSA tissue scanner with ImageScope software v12.4.3.5008 (Aperio; Sausalito, CA, USA).

### 4.9. Measurement of MMP Activity

The conditioned media were collected from acinar-to-ductal metaplasia assay at day 3–5 and concentrated using Amicon^®^ ultra centrifugal filters with a 30 kDa cut-off. The concentrated conditioned media were subject to MMP2 activity assay using an MMP2 activity assay kit (AnaSpec; Fremont, CA, USA) and MMP3 activity assay using an MMP3 activity assay kit (Abcam), respectively, according to the manufacturer’s instructions.

### 4.10. Measurement of Cellular ROS Levels

The primary pancreatic acinar cells in each condition were labeled with 20 µM H_2_DFFDA in complete media at 37 °C for 20 min. Cells were then washed 3 times with pre-warmed Live Cell Imaging Solution (Thermo Fisher Scientific). After the final wash, the cells were placed in a sterile black 96-well plate with a clear bottom, and cellular ROS levels were measured in Live Cell Imaging Solution using the Synergy H1 Hybrid Microplate Reader (Agilent Technologies) at 485/528 (Ex/Em) nm in a time course of a total of 12 h.

### 4.11. Quantification and Statistical Analysis

All cell biological and biochemical experiments were carried out at least three times. Each repeat experiment was performed using pancreas tissue from a different individual mouse. For animal experiments, at least three pancreas samples were subject to quantification analysis. Five randomized fields from each sample were utilized for quantification. Data are presented as means ± SE. *p*-values were acquired with the Student’s *t* test for comparing 2 sets of data using Prism 7 (GraphPad Software (San Diego, CA, USA)). For more than 2 sets of data, one-way ANOVA analysis along with multiple comparisons was carried out using Prism 7 (GradPad Software). *p* < 0.05 is considered statistically significant.

## Figures and Tables

**Figure 1 ijms-25-04726-f001:**
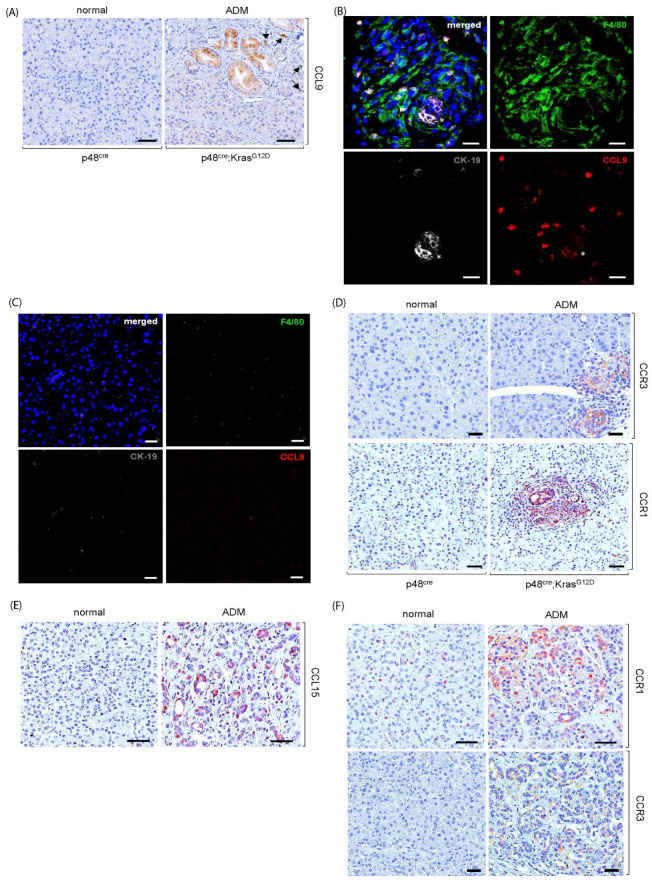
Elevated expression of CCL9 and its receptors in the ADM regions of pancreas tissue samples. (**A**) Pancreas tissues of p48^cre^:Kras^G12D^ mice and their matching littermate p48^cre^ mice were immunostained with antibodies of CCL9. Scale bar: 50 µm. Arrow: CCL9-positive immune cells. (**B**) The ADM area of p48^cre^:Kras^G12D^ mouse pancreas was fluorescently immunostained with antibodies of F4/80 (green, macrophage marker), CK-19 (grey, ductal marker) and CCL9 (Red). The cells were visualized by DAPI labeling (blue). Asterisk indicates the duct-like structures derived from acini (where ADM occurred). Scale bar: 25 µm. (**C**) Exactly that same as (**B**), except using the pancreas tissue of p48^cre^ mouse. (**D**) Pancreas tissues of p48^cre^:Kras^G12D^ mice and their matching littermate p48^cre^ mice were immunostained with antibodies of CCR1 and CCR3. Scale bar: 50 µm. (**E**,**F**) Human pancreas tissue samples that contain ADM areas and normal pancreas tissues were immunostained with antibodies of CCL15 (**E**), CCR1 and CCR3 (**F**). Scale bar: 50 µm.

**Figure 2 ijms-25-04726-f002:**
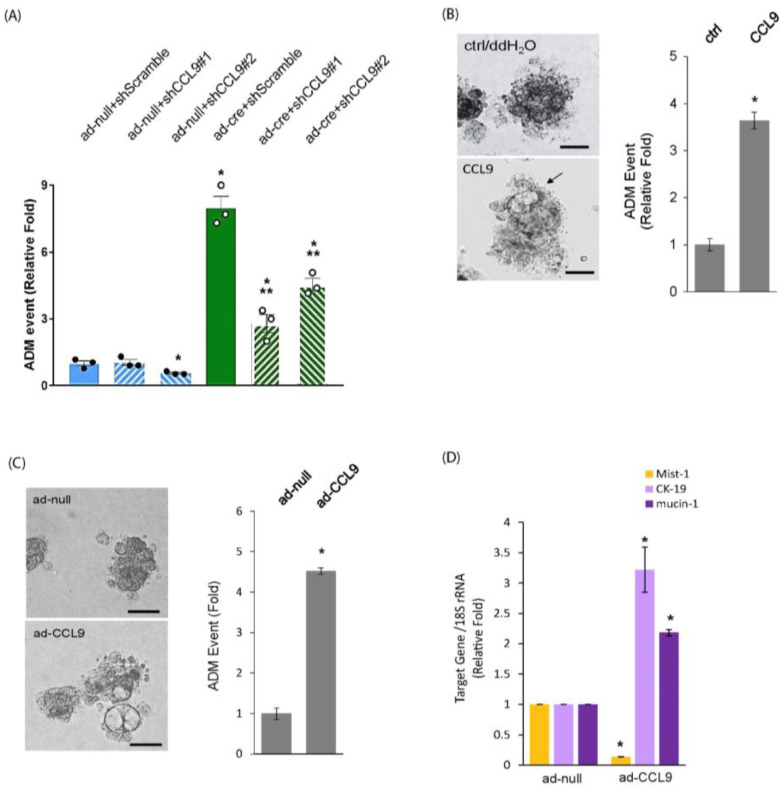
CCL9 was required for Kras^G12D^-induced pancreatic ADM and capable of driving ADM of the pancreas in 3D organoid culture system. (**A**) The isolated primary pancreatic acini of LSL-Kras^G12D^ mouse were infected with adenovirus (ad-cre or ad-null/ctrl) and lentivirus (shCCL9 #1, shCCL9 #2 or shScramble/ctrl) and subject to ADM analysis. At the endpoint, the number of ADM structures under each condition was counted and analyzed as ADM events. *: *p* < 0.05 as compared to ad-null+ shScramble; **: *p* < 0.05 as compared to ad-cre + shScramble. (**B**,**C**) Primary pancreatic acini from wildtype mouse were treated with either recombinant CCL9 or ddH_2_O/ctrl (**B**) or infected with ad-CCL9 adenovirus or ad-null adenovirus (**C**) followed by the acinar-to-metaplasia assay in 3D as described in the Section 4 At the endpoint, the number of ductal structures were counted and analyzed as ADM events. Cells were also photographed to document the ADM event as shown. Arrow: ductal structure. *: *p* < 0.05 as compared to control. Scale bar: 50 µm. (**D**) At the endpoint, cells from (**C**) were isolated for and subject to RNA extraction, followed by real-time qRT-PCR for assessing Mist-1, CK-19 and mucin-1 transcripts. *: *p* < 0.05 as compared to ad-null.

**Figure 3 ijms-25-04726-f003:**
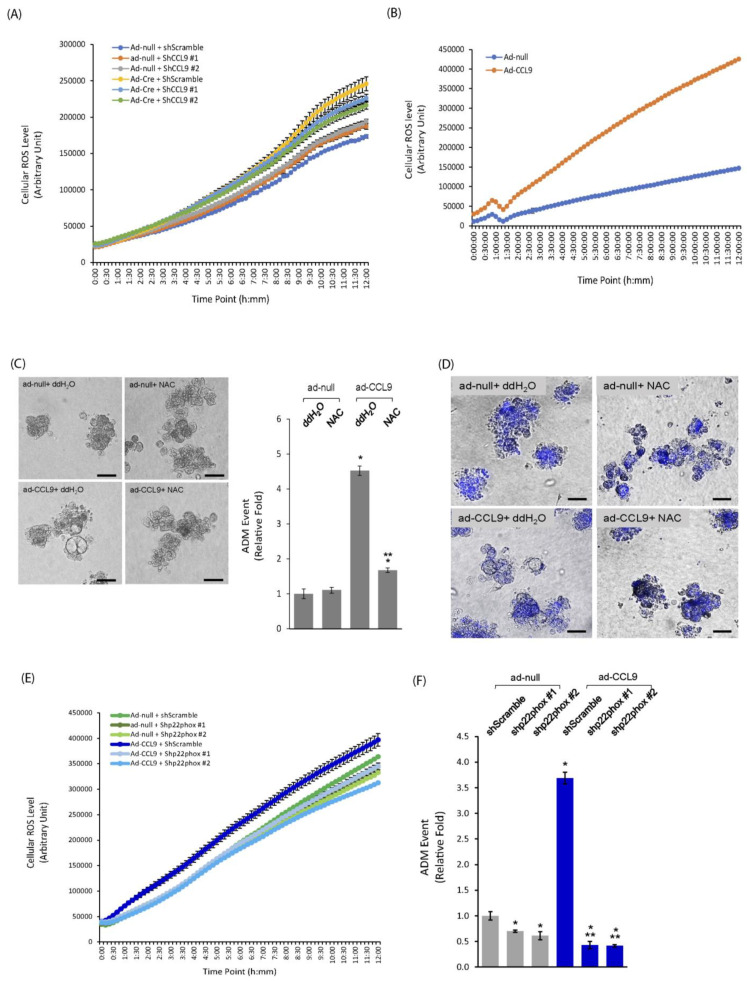
CCL9 increased ROS levels to regulate pancreatic ADM. (**A**) Primary acini isolated from LSL-Kras^G12D^ mouse pancreas were infected with adenovirus (ad-null or ad-cre) and lentivirus (shScramble/ctrl, shCCL9 #1 or shCCL9 #2), as described in Materials and Methods. Forty-eight hours post-infection, the cells were evaluated for their cellular ROS levels. (**B**) Similar to (**A**), primary pancreatic acini isolated from wildtype mouse were infected with ad-null or ad-CCL9 adenovirus and subject to the detection of their intracellular ROS levels. (**C**) Primary pancreatic acini from wildtype mouse infected with ad-null or ad-CCL9 adenovirus were subject to ADM analysis in the presence or absence of NAC. At the endpoint, the number of ADM structures under each condition was counted and analyzed as ADM events. *: *p* < 0.05 as compared to ad-null+ ddH_2_O/ctrl; **: *p* < 0.05 as compared to ad-CCL9+ ddH_2_O. Scale bar: 50 µm. (**D**) As described in (**C**), at the endpoint, cells embedded in collagen were incubated with Hoechst 33342 to determine their viability. Images shown are merged with the bright-field channel and Hoechst channel. Scale bar: 50 µm. (**E**) Similar to (**B**), WT mouse pancreatic acini infected with ad-null or ad-CCL9 in the presence of lentivirus of shScramble, shp22phox #1 or shp22phox #2 were evaluated for their cellular ROS levels. (**F**) The acini as described in (**E**) were subject to ADM assay. *: *p* < 0.05 as compared to ad-null/ctrl+ shScramble; **: *p* < 0.05 as compared to ad-CCL9+ shScramble.

**Figure 4 ijms-25-04726-f004:**
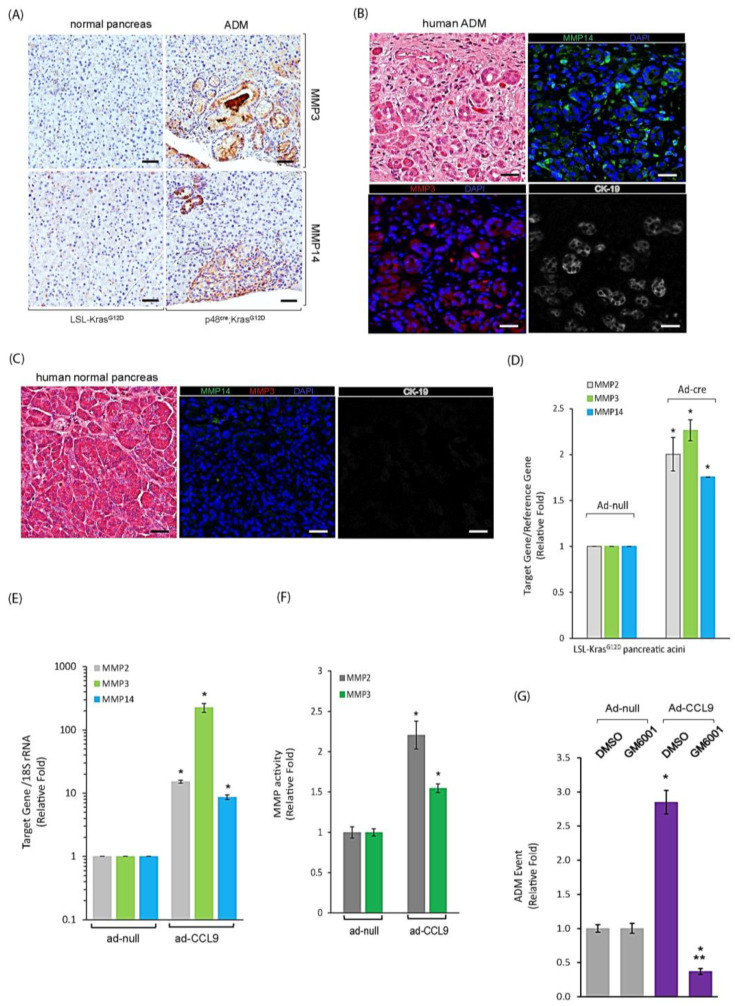
MMPs as downstream mediators of Kras^G12D^-CCL9 signaling to promote ADM of the pancreas. (**A**) Pancreas tissues of p48^cre^:Kras^G12D^ mice and their matching littermate LSL-Kras^G12D^ mice were immunostained with antibodies of MMP3 and MMP14. Scale bar: 50 µm. (**B**,**C**) Human pancreas tissue samples containing ADM regions (**B**) or human normal pancreas (**C**) were stained with H&E and fluorescently immunostained with antibodies of MMP14 (green), MMP3 (red) and CK-19 (grey). DAPI labeling was used to visualize all cells. Scale bar: 50 µm. (**D**,**E**) Primary pancreatic acini isolated from either LSL-Kras^G12D^ mouse (**D**) or wildtype mouse (**E**) were infected with adenovirus as indicated and then embedded in collagen. At the endpoint, cells were isolated for RNA extraction and subject to real-time qRT-PCR for mRNA levels of MMP2, MMP3 and MMP14. *: *p* < 0.05 as compared to ad-null. (**F**) Pancreatic acini isolated from wildtype mouse were infected with ad-null or ad-CCL9 and then embedded in collagen in 3D. The conditioned media collected from the 3D organoid culture were subject to MMP activity assay. *: *p* < 0.05 as compared to ad-null. (**G**) Similar to (**E**), primary pancreatic acini infected with adenovirus of ad-null or ad-CCL9, treated with or without 2 µM GM6001, were subject to ADM assay. The number of ADM structures under each condition was quantified and presented as ADM events. *: *p* < 0.05 as compared to ad-null+ DMSO/ctrl; **: *p* < 0.05 as compared to ad-CCL9+ DMSO.

**Figure 5 ijms-25-04726-f005:**
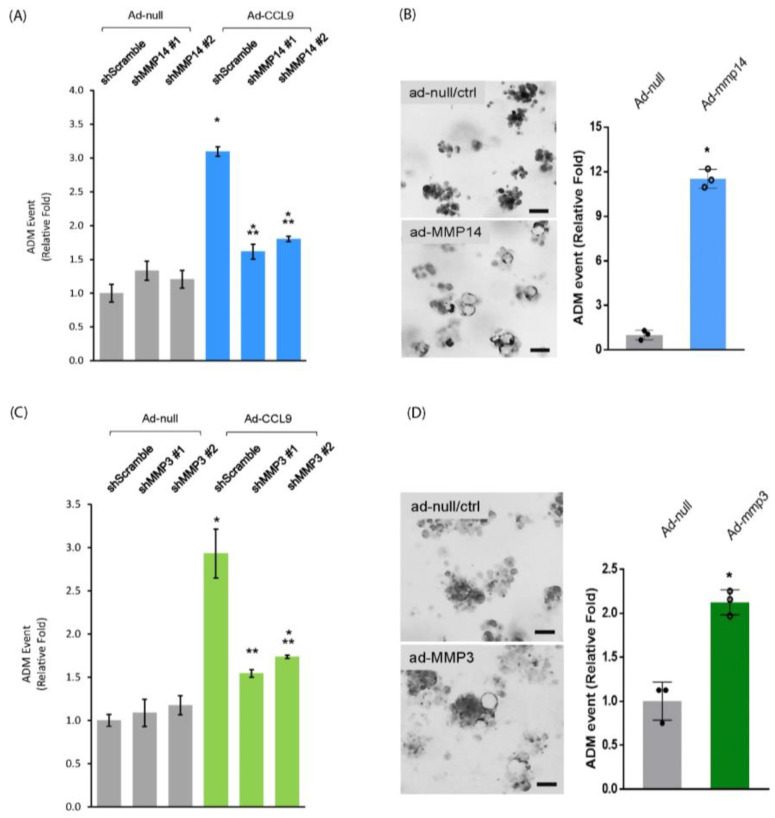
CCL9 signaled through MMP14 and MMP3 to modulate pancreatic ADM. (**A**) Primary pancreatic acini isolated from wildtype mouse were infected with lentivirus of shScramble or shMMP14 along with adenovirus of ad-null or ad-CCL9, followed by embedding in collagen. At the endpoint, the number of ductal structures were counted, quantified and analyzed as ADM events. *: *p* < 0.05 as compared to shScramble + ad-null; **: *p* < 0.05 as compared to shScramble + ad-CCL9. (**B**) Similar to (**A**), primary pancreatic acini infected with either ad-null or ad-MMP14 adenovirus were subject to ADM assay. *: *p* < 0.05 as compared to ad-null. Scale bar: 50 µm. (**C**) Similar to (**A**), primary acini from wildtype mouse pancreas were infected with lentivirus of shScramble or shMMP3 in the presence of ad-null or ad-CCL9 adenovirus. These cells were then subject to ADM assay. The graph presents the ADM events. *: *p* < 0.05 as compared to shScramble + ad-null; **: *p* < 0.05 as compared to shScramble + ad-CCL9. (**D**) Similar to (**B**), primary murine pancreatic acini infected with ad-null or ad-MMP3 adenovirus were subject to ADM assay. *: *p* < 0.05 as compared to ad-null. Scale bar: 50 µm.

**Figure 6 ijms-25-04726-f006:**
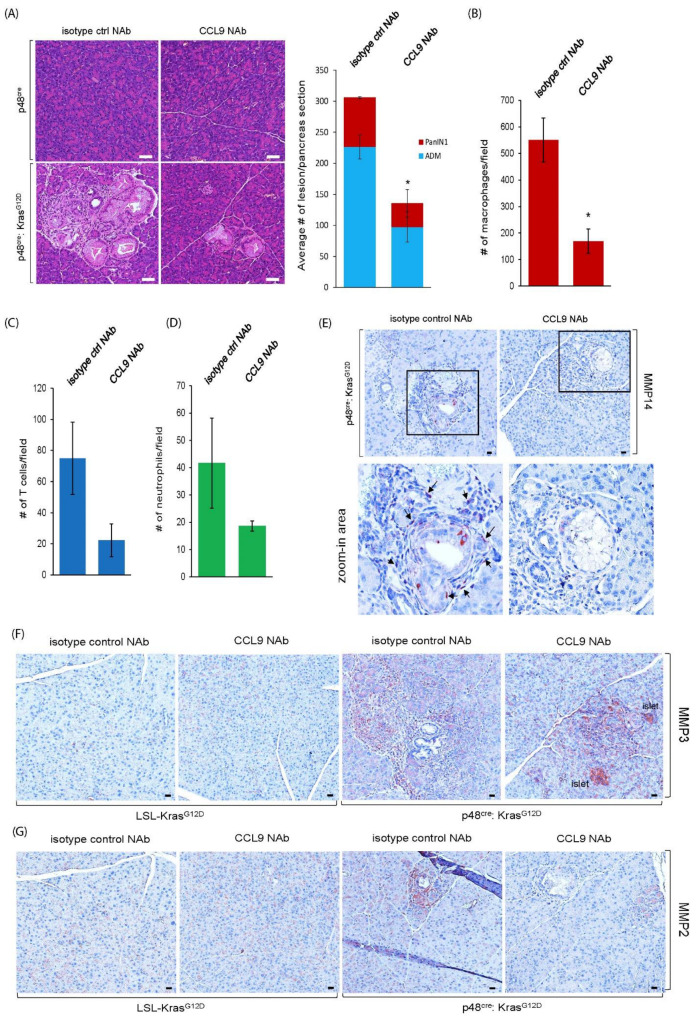
Blockade of CCL9 reduced Kras^G12D^-induced pancreatic ADM, filtrating macrophages and MMPs in vivo. (**A**) p48^cre^:Kras^G12D^ mice and age-matched littermate p48^cre^ mice at 3 weeks old were treated with either isotype control neutralizing antibody (NAb) or CCL9 Nab every other day for 5 weeks, as described in the Section 4. Pancreas tissues at the endpoint were harvested and stained with H&E to visualize abnormal structures. The number of pancreatic ADM structures and PanIN lesions per pancreas section were analyzed and quantified. Scale bar: 50 µm. *: *p* < 0.05 as compared to isotype control NAb. (**B**–**D**) The pancreas tissue samples from (**A**) were immunostained with antibodies of F4/80 (macrophage marker), CD3 (T cell marker) or Ly6B.2 (neutrophil marker) to evaluate the infiltrated immune cells surrounding the ADM regions. Image J was used to quantify the density of infiltrating immune cells including macrophages (**B**), T cells (**C**) and neutrophils (**D**) in the ADM regions of p48^cre^:Kras^G12D^ mouse pancreas according to their immunohistochemistry results (see Appendix A). *: *p* < 0.05 as compared to isotype control NAb. (**E**–**G**) Pancreas tissue samples from (**A**) were immunostained with antibodies of MMP4 (**E**), MMP3 (**F**) and MMP 2 (**G**). Arrow: MMP14-positive infiltrating immune cells surrounding the ADM region. Scale bar: 50 µm.

**Figure 7 ijms-25-04726-f007:**
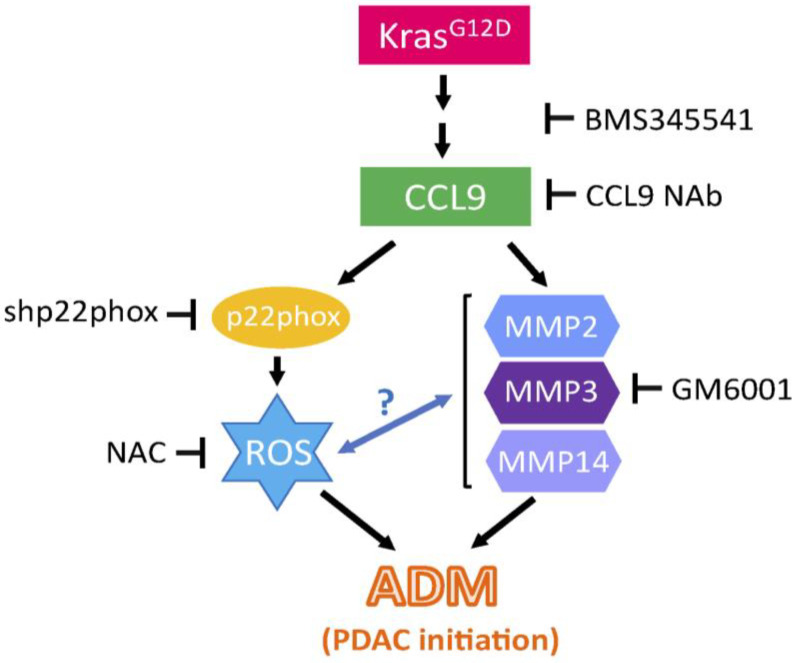
Summarized findings on how Kras^G12D^ potentiates ADM via CCL9 and other downstream molecules to initiate PDAC. This diagram is depicted based on our findings from this study to demonstrate the mechanism that Kras^G12D^ utilizes to initiate PDAC. ROS: reactive oxygen species; NAC: N-Acetyl Cysteine; MMP: matrix metalloproteinase; NAb: neutralizing antibody; ADM: acinar-to-ductal metaplasia. Question mark: requires further scientific evidence.

## Data Availability

The data generated in this study are available within the article and Appendix A.

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
