# Peer review of "Cytokine CCL9 Mediates Oncogenic KRAS-Induced Pancreatic Acinar-to-Ductal Metaplasia by Promoting Reactive Oxygen Species and Metalloproteinases"

_ijms, 2024, doi:10.3390/ijms25094726_

Round 1
Reviewer 1 Report (Previous Reviewer 2)
Comments and Suggestions for Authors
accept it
Comments on the Quality of English Languageaccept it
Author Response
Please see the attachment for the detailed point-to-point response.

Reviewer 2 Report (New Reviewer)
Comments and Suggestions for Authors
The manuscript investigated the role played by CCL9 in the KRAS-associated pancreatic acinar-to ductal metaplasia (ADM)using LSL-KRASG12D mice, primary cultures of cell pancreatic acini to induce ADM, and human pancreas tissues containing ADM areas.
The possible involvement of increased ROS production and MMP expression has been also investigated.
The experimental design is correctly structured and involves confirmation of the results through silencing technique or exogenous administration of CCL9. Anyway, some aspects need to be analyzed in more depth.
1) The authors conclude that "cytokine CCL9 as a new downstream target of 315 KrasG12D to modulate pancreatic ADM during PDAC initiation", but no information is reported about the signal transduction pathway(s) starting from KRAS mutated and leading to the increased expression of CCL9. In fact, it is well known that KRAS can modulate gene transcription through different pathways (RAF-MKK, PI3K/AKT, Rol/NFkB). Experiments aiming to identify the pathway responsible for the increased CCL9 expression have to be performed.
Since as reported by the author "there are no inhibitors for oncogenic KRAS mutations", this aspect could be very important to identify new potential targets with the aim of reducing the KRAS-induced ADM and the subsequent PDAC development/progression.
2) The authors conclude that the increased CCL9 expression favors the ADM onset through a major production of ROS and that this process is reverted by NAC administration. In the Discussion they cite their previous study and other authors to explain that the CCL9-induced ROS increase can be mediated by its effect on p22phox subunit of NADPH oxidase.
A direct evidence of the mechanism responsible for the CCL9-mediated increase of ROS in the present experimental model needs to be added in the manuscript.
For the above reasons, the Scheme proposed by the authors in the Fig. 7 is currently speculative and needs to be completed after characterizing the mechanisms underlying the observed effects.
Minor comments:
1) In the Results section general statements have to be avoided (lines 79-81; 116-117; 161) mainly when they are already present in the Introduction or in the Discussion.
2) In the figure legends the description of methodologies used needs to be eliminated since they are already reported in Materials & Methods, this avoiding the legends being too long.
3) The discussion has to be more focused on the manuscript results without mentioning general knowledge about KRAS or ADM PDAC.
4) Several statements need to be revised from a conceptual point of view. For example: a) Since there are no inhibitors for oncogenic Kras mutations; b) Acinar-to-ductal metaplasia (ADM) is the process that oncogenic KrasG12D utilizes to 79 convert quiescent pancreatic acini to a proliferative duct-like phenotype, which initiates 80 pancreatic ductal adenocarcinoma (PDAC) c) These results suggested two mechanisms utilized by KrasG12D-expressed pancreatic acini to initiate PDAC through upregulation of ADM.
Comments on the Quality of English Language
The quality of English language is fine
Author Response
Please see the attachment for the detailed point-to-point response.

Reviewer 3 Report (New Reviewer)
Comments and Suggestions for Authors
The manuscript offers valuable insights into the molecular mechanisms driving oncogenic KrasG12D-induced pancreatic acinar-to-ductal metaplasia (ADM) and the initiation of pancreatic ductal adenocarcinoma (PDAC). Specifically, the authors identify CCL9 as a novel driver of ADM and a downstream target of KrasG12D signaling during PDAC initiation. They demonstrate that CCL9 induces the transdifferentiation of pancreatic acini into a duct-like phenotype through the mediation of reactive oxygen species (ROS) and matrix metalloproteinase (MMP) proteins. The presentation of data is clear and well-organized, contributing to the manuscript's strength. However, several points merit attention for clarity and to address potential gaps in the research:
- 1. The inclusion of results regarding the efficiency of CCL9 shRNA would bolster the credibility of the findings.
- 2. While the authors demonstrate that overexpression of CCL9 increases MMP protein expression in Kras-induced ADM, it remains unclear how MMP expression is affected when CCL9 is knocked down. This aspect warrants further investigation to provide a comprehensive understanding of the relationship between CCL9 and MMP proteins.
- 3. To enhance the robustness of the study, it would be beneficial to treat p48cre/lsl-KrasG12D mice with N-acetyl-L-cysteine (NAC) and/or GM6001, as these interventions could provide additional insights into the role of ROS and MMP proteins in CCL9-mediated ADM.
- 4. While the discussion effectively contextualizes the results within the existing literature, further exploration of the study's limitations would strengthen this section. Addressing potential constraints and outlining future research directions would enhance the overall discussion of the manuscript.
Author Response
Please see the attachment for the detailed point-to-point response.

Round 2
Reviewer 2 Report (New Reviewer)
Comments and Suggestions for Authors
The authors declare that they disagree with all the referee's comments/suggestions. For this reason, only few changes in the revised version of the manuscript have been made and they did not significantly improve it.
Author Response
Please see the attachment for the point-to-point response. Thank you!

This manuscript is a resubmission of an earlier submission. The following is a list of the peer review reports and author responses from that submission.
Round 1
Reviewer 1 Report
Comments and Suggestions for Authors
1. In this manuscript, the authors have investigated the role of Cytokine CCL9 in mediating oncogenic KRAS-induced pancreatic acinar-to-ductal metaplasia. It is noted that CCL9 is also known by various names such as macrophage inflammatory protein-1 gamma (MIP-1γ), macrophage inflammatory protein-related protein-2 (MRP-2), and CCF18 in rodents. The manuscript should include a description of these alternative names to provide a comprehensive understanding of CCL9.
2. The manuscript should discuss potential mechanisms underlying how CCL9 induces the upregulation of intracellular levels of reactive oxygen species (ROS). This discussion would enhance the clarity of the findings and contribute to a better understanding of the molecular processes involved.
3. It would be valuable if the authors could explore any clinical evidence linking elevated levels of CCL9 with pancreatitis or pancreatic cancer. Integrating relevant clinical studies into the manuscript would provide a more comprehensive context for the clinical implications of the reported findings.
4. The authors are encouraged to consider running gelatin gels for the detection of active forms of MMP14, MMP3, and MMP2. This additional experimental approach would strengthen the robustness of the findings and offer a more comprehensive analysis of MMP activity in the context of pancreatic acinar-to-ductal metaplasia.
Reviewer 2 Report
Comments and Suggestions for Authors
· The authors utilized 3D organoid culture system to evaluate if the increased levels of MMP proteins in ADM were due to an upregulation of MMP gene expressions by KrasG12D and/or CCL9. Since MMPS are also involved in epithelial-mesenchymal transition (EMT) the authors should evaluate the levels of EMT markers (such as e-cadherin, vimentin, snail1, twist1) in shScramble/control, shCCL9 #1 or shCCL9 #2 samples
· Recently, Delle Cave and colleagues demonstrated that the overexpression of LAMC2 improves the tumorigenic potential of the PDAC cells both in vitro and in vivo, together with the overexpression of MMPs (doi: 10.1186/s13046-022-02516-w). As the expression of LAMC2 in ADM and its connection with CCl9 is unknown, the authors must evaluate the expression of LAMC2 in ADM regions of p48cre:KrasG12D mouse pancreas via immuno-histochemistry.
Comments on the Quality of English LanguageGood